# Exploring First Responders’ Use and Perceptions on Continuous Health and Environmental Monitoring

**DOI:** 10.3390/ijerph20064787

**Published:** 2023-03-08

**Authors:** Jacob Grothe, Sarah Tucker, Anthony Blake, Chandran Achutan, Sharon Medcalf, Troy Suwondo, Ann Fruhling, Aaron Yoder

**Affiliations:** 1College of Public Health, University of Nebraska Medical Center, Omaha, NE 68198, USA; 2College of Information Science & Technology, University of Nebraska-Omaha, Omaha, NE 68182, USA

**Keywords:** first responders, physiological, safety, sensor monitoring, survey, attitude, behavior

## Abstract

First responders lose their lives in the line of duty each year, and many of these deaths result from strenuous physical exertion and exposure to harmful environmental agents. Continuous health monitoring may detect diseases and alert the first responder when vital signs are reaching critical levels. However, continuous monitoring must be acceptable to first responders. The purpose of this study was to discover first responders’ current use of wearable technology, their perceptions of what health and environmental indicators should be monitored, and who should be permitted to monitor them. The survey was sent to 645 first responders employed by 24 local fire department stations. A total of 115 (17.8%) first responders answered the survey and 112 were used for analysis. Results found first responders perceived a need for health and environmental monitoring. The health and environmental indicators that respondents perceived as most important for monitoring in the field were heart rate (98.2%) and carbon monoxide (100%), respectively. Overall, using and wearing monitoring devices was not age-dependent and health and environmental concerns were important for first responders at any stage of their career. However, current wearable technology does not seem to be a viable solution for first responders due to device expense and durability issues.

## 1. Introduction

Being a first responder requires strenuous physical exertion in addition to coping with environmental hazards. This occupation is physically, mentally, and emotionally demanding. Each year first responders experience high-injury rates from accidents, musculoskeletal health complications, and sudden premature deaths. Among these occupational health events, coronary heart disease (CHD) is the leading cause of death for first responders, and nearly half of the CHD events occur during first responder activity [1].

There are many different risks for first responders. Between the years 1994 and 2004, nearly 368 United States (U.S.) career first responders lost their lives while on duty, with 39% of those deaths resulting from myocardial infarction (MI) [1]. Cancer has also been identified as a health concern for first responders. A study identifying 3996 male firefighters found them to have significantly elevated risks for melanoma, multiple myeloma, acute myeloid leukemia, and cancer of the esophagus, prostate, brain, and kidney [2]. Compared to the total number of emergencies to which first responders respond, fires are relatively rare and yet most duty-related CHD events happen during fire suppression responses. Fire suppression increases the heart rate and blood pressure levels to maximal age-predicted levels during the event and exposes first responders to extreme temperatures. However, not all MI events result in death; for at least every one fatal MI, seventeen non-fatal MI events occur in the United States [1,3].

Common contributors to the morbidity and mortality from MI and CHD are the long shifts, psychological stressors, the heavy weight of their personal protective equipment (PPE), the amount of exposure to smoke, and chemicals, as well as the high levels of physical exertion [4]. These stressors are some of the leading explanations for mortality and morbidity in first responders with underlying left-ventricle hypertrophy or CHD [4]. Prolonged periods of high strenuous activity can lead to a decrease in myocardial function, causing a decrease in the systolic and diastolic functions of the heart and leading to potentially fatal outcomes (i.e., MI) [5].

Another concern is the increase in the prevalence of obesity and overweight among firefighters, especially in the U.S. population [6]. Poston et al. tested first responders’ body mass index (BMI) in addition to their body fat percentage to determine if they are genuinely classified as overweight or obese (2011) [6]. Evidence has shown obese and overweight individuals have a higher risk of developing cardiovascular disease [6]. From the study, 667 first responders were measured, and the overall average BMI was greater than 25 [6]. Additionally, 32.7% who were classified as overweight using BMI testing were obese based on body fat percentages [6]. Hypertension among first responders is approximately 20–30%, with 50% having prehypertension, and it is expected to increase due to the obesity epidemic in the United States [3,5].

Hypertension causes vascular changes to the cardiovascular system and can lead to complications such as left ventricle hypertrophy and atherosclerosis. Hypertension can also increase the risk of sudden onset events such as MI, stroke, and heart failure [7]. Left-ventricle hypertrophy was documented during autopsies in nearly 56% of first responders who had a fatal occurrence of heart disease, which suggests that uncontrolled hypertension may be present in a majority of first responders [3]. The combination of risk factors mentioned above, and physiological responses, such as increased heart rate, blood pressure, and body temperature that occur during first responder activity, increase the likelihood of cardiovascular disease (CVD).

Multiple studies have demonstrated that first responders’ physiological values increase significantly during strenuous first responder physical activities. For instance, a study on career and volunteer firefighters between the age of 18–24 with no history of CVD, smoking, or diabetes tested the effects of PPE weight during physical simulations. The results demonstrated an increase in both heart and blood pressure during activities wearing and not wearing PPE. The study, however, showed a larger increase in heart rate and blood pressure while wearing PPE [8]. Another study observed a 3 h live-fire training exercise with 40 male firefighters with a mean age of 27.4 years and found the heart rate to increase, ranging between 159 to 213 bpm during fire suppression [5]. These findings are consistent with other findings analyzing heart rate or blood pressure. Regardless of age, body weight, type of physical stimulation, or the timing of each activity, each study showed an increase in heart rate to near to max age-predicted values [1,8,9,10,11,12,13].

Only a few studies collected data on the first responders during recovery between short bouts of high physical exertion. These studies found similar results, namely, that recovery in between each session of work is not adequate for a full recovery of the first responder [8,11,13]. For instance, 16 male firefighters with a mean age of 29 years and a mean weight of 91.2 kg had their heart rate, membrane temperature, blood glucose levels, and blood lactate recorded before the drill, after 8 min and 16 min within the task, and 10 min after recovery. In addition, they were tested in two different temperature environments: one being neutral temperature (13.7 °C) and the other being 89.6 °C. The heart rate within the 89.6 °C environment increased the heart rate to age-predicted values and remained high (mean of 137.33 ± 8.19) even after 10 min of recovery [13].

Another study reported on nine firefighters with a mean age of 32.8 years and analyzed a 3 h live-fire training exercise and its effects on core temperature and heart rate. The 3 h event had four work cycles lasting between 15 and 30 min with rest periods lasting between 20 and 40 min. This work-rest ratio is seen in real live firefighting events. The heart rate continued to increase after each activity, peaking at 188 bpm. Additionally, body temperature increased after each event, peaking at 38.72 °C [11]. Even during the 20–40 min recovery time, the body does not fully recover after each event because the heart rate does not return to near resting levels before the next work cycle [11].

Heart rate and blood pressure are not the only physiological changes that can increase the risk of CVD. Hyperthermia, the increase in body temperature, may trigger sudden cardiac events. A systematic review evaluating studies that recorded heart rate and temperature reported that body temperature was shown to increase on average by 0.9 °C during firefighting activities. The increase in body temperature was caused by exposure to heat while wearing a self-contained breathing apparatus (SCBA) and PPE, which do not favor thermoregulation. PPE has been shown to increase skin temperature and internal temperature as well as decrease heat transfer due to the limited permeability of water vapor [4].

An individual who experiences heat stress causes a significant strain on the cardiovascular system. Heat stress has been revealed to cause a significant reduction in blood volume, especially in organs such as the heart, liver, and spleen by nearly 14% or higher [14]. In addition to blood volume reduction, heat stress causes a reduction in ventricular filling pressure and stroke volume, which is accompanied by an increase in heart rate. This results in a significant reduction in cardiac output [14]. In combination with whole-body exercise, heat stress can cause cardiovascular strain due to the reduction in cardiac output, stroke volume, and even blood flow to the muscles and brain [14].

### 1.1. Wearable Health Monitoring Devices

Many of these physiological changes that occur during these events can be measured by continuous wearable health-monitoring devices allowing first responders to assess their occupational health better. These systems can provide real-time feedback on one’s health information. Additionally, they can alert the first responders when acute life-threatening conditions are imminent. Continuously tracking first responders’ health physiological changes may also lead to an early diagnosis of disease, as well as early treatment, ultimately improving their quality of life [15,16]. Continuous monitoring can provide more information on a person’s health, such as arrhythmias, which can be an indicator of heart disease [15]. Additionally, the measurement of multiple vital signs, such as blood pressure, heart rate, respiratory rate, and blood oxygen saturation levels, can bring greater insight into the pathophysiology of disease and new indications of physiological markers of disease status [15].

### 1.2. Vital Signs

Heart rate, blood pressure, respiratory rate, blood oxygen saturation, and body temperature are five vital signs that can be considered essential in providing accurate information about an individual’s health [17]. Electrocardiograms (ECGs) are a widely used biometric instrument that evaluates cardiac rhythm through electrical analysis. ECG can also serve as a diagnostic tool within healthcare and can predict acute MI and other coronary events, such as atherosclerosis, CHD, tachycardia, and bradycardia [17]. Respiratory Rate (RR) can determine cases of distress that can cause hypoxia. Further, respiratory rate can detect other diseases such as chronic pulmonary disease, which are risk factors for the onset of CVD and acute MI. When low percentages are present in blood oxygen saturation levels during an event such as hypoxia, this causes insufficient oxygen (O_2_) supply to the body. As mentioned previously, an increase in body temperature, which can be affected by extreme heat, places first responders at risk for CVD [4].

### 1.3. Environmental Factors

Measuring the environment also has its importance. Many exposures within the environment can potentially lead to diseases, such as radon, causing lung cancer, arsenic causing, cancer of several organs, and particulate matter, causing cardiorespiratory diseases [18,19]. Furthermore, according to the Agency for Toxic Substances and Disease Registry (ATSDR), ammonia can lead to airway destruction causing respiratory failure [20]. Hydrogen sulfide (H_2_S) in high concentrations quickly leads to death [21]. Lastly, carbon monoxide (CO) can lead to cardiovascular diseases or death if exposure is high [22]. First responders are potentially exposed to these toxic chemicals during fire suppression because modern building materials and furnishing are becoming more synthetic and, when combusted, release these toxic gas byproducts [23,24]. Other environmental exposures include noise. According to OSHA, high levels of noise can cause permanent hearing loss and loud noise can create physical and psychological stress, reduce productivity, and interfere with communication and concentration which all are important factors to first responders [25].

Each year numerous first responders lose their lives due to MI or CVD; these numbers could be reduced if first responders had more preventive measures in place. Continuous monitoring has the potential to detect diseases early by providing information about the health of the firefighters during emergencies [26]. Furthermore, information regarding the environment can provide firefighters with essential information when arriving at an emergency event. Due to the health conditions mentioned previously, there are apparent reasons for first responders to be monitored; however, there is limited research exploring first responder’s current use of health monitoring technology, which physiological or environmental indicators are perceived as important to them, and/or who should monitor their information during emergencies. Therefore, the study objectives for this project are as follows:Determine how many of the first responders are currently using wearable technology. This gives insight into what health indicators are already monitored and the kind of systems being used.Determine which health and environmental indicators are perceived to be important to first responders. There are many different indicators, and identifying which ones are most important can help a monitoring system focus on those indicators.Determine who should monitor the first responders’ health status and environmental status within the field.Examine if age influences the responses to health monitoring.Examine if the role (e.g., special operations) influences the responses to health monitoring.Examine if age influences the responses to environmental monitoring.Examine if the role (e.g., special operations) influences the responses to environmental monitoring.

## 2. Materials and Methods

### 2.1. Sampling of Participants

The survey was administered in 2019 to first responders who worked in the local area fire departments. First responders from all twenty-four fire departments were eligible to participate in the study, and there were no inclusion/exclusion criteria for the study. Before being administered, the survey was approved by the Institutional Review Board (IRB) (IRB # 691-17-EX) of the University of Nebraska Medical Center (Omaha, NE, USA).

### 2.2. Measurements

A 16-question survey was developed to answer the research questions. These health and environmental key variables were identified in our previous study [27]. Three demographic questions assessed the location of the participant’s fire station by zip code, whether the participant was part of special operations, and what year that participant was born. In general, special operations first responders often work in emergency medical care, technical rescue, hazardous materials mitigation, etc. Four questions measured whether the participant ever used wearable technology, and if the participant answered yes to ever using wearable technology, then they were asked if they use wearable technology and how confident they were when operating wearable technology. If the participant answered no to ever using wearable technology, then they were asked why they do not use wearable technology. Three questions evaluated which health information would be useful to monitor when working in the field and who should monitor standardized emergency management system (SEMS) operator, which is someone monitoring a system that displays vital information of the first responder, and the first responder themselves. Lastly, two questions asked the participant if any additional types of health or environmental information should be monitored that were not listed as an option within the survey.

### 2.3. Statistical Analysis

Questionnaires with incomplete information were excluded from the analysis. Using SAS Software (version 9.4; SAS Institute, Inc., Carly, NC, USA), the association between non-special operations and special operations and current use in technology, the types of health and environmental information that should be monitored, and who should monitor that information were determined using univariate chi-square analysis. A univariate chi-square analysis was also used to assess if the current use of technology, the types of health and environmental information that should be monitored, and who should monitor that information was associated with the age of the individual. Age was categorized into two groups based on the mean age of the respondents. The age was not normally distributed, so we selected 42 based on the distribution where it naturally broke at the mean age of 42. Respondents were either placed in a group below the age of 42, or respondents were placed in a group equal to or greater than the mean age of 42 years. The final sample size for analysis to examine the association between non-special operations and special operations as well as an association between the age of the respondents was *n* = 112. Responses with low frequencies for health information variables were re-coded and categorized dichotomously. The “no” and “don’t” know responses were combined to “no” and the yes responses remained “yes” since we were only interested in who desired a health variable to be monitored. Similarly, we re-coded environmental factor variables dichotomously. We combined the responses for who should monitor that information for the same reason. The new categories for who should monitor that information were “SEMS operator and myself,” “SEMS operator only,” “myself,” and “other”. Finally, for the wearable technology use variable concerning how confident they were in using wearable technology we combined the “very confident” and “somewhat confident” responses into one response category renamed “low confidence” and the extremely confident response was renamed “high confidence”.

## 3. Results

### 3.1. Study Subject Characteristics

Responses were collected from 115 out of the 645 first responders in the local fire department. From the 115 responses, only two respondents were omitted from the analysis due to missing data. Out of the 113 who completed the questionnaire, 112 answered the question, “Whether they are part of the special operations group?”. Based on that question, there were 78 (70%) who were not part of the special operations group, and 34 (30%) who were part of the special operations group. Individuals had an average age of 42 ± 7.6 years standard deviation (SD). Out of those, 45 (40.18%) participants were under the age of 42, and 67 (59.82%) participants were greater than or equal to 42 years of age.

### 3.2. Current or Past Use of Technology and Confidence

Among the 112 responses to the question of whether they have ever used wearable technology (i.e., Fitbit, Smartwatch), 53 (47.32%) selected yes. Among the 53 respondents, 31 (58.49%) of the respondents selected that they were currently using wearable technology. In addition, 16 of the 31 (51.61%) reported high confidence in their ability to operate wearable technology and 15 of the 31 (48.39%) reported low confidence. Table 1 presents the results of a univariate analysis that assessed the association between special operations and non-special operations and wearable technology. Based on Table 1, there was no significant association between ever using wearable technology (*p* = 0.11), the current use of wearable technology (*p* = 0.86), and the confidence in using wearable technology (*p* = 0.11). Table 2 reveals the results of the univariate association between the age groups (Age ≥ 42 or Age < 42) and wearable technology. Similar results were observed: there was no significant difference in age and between ever using wearable technology (*p* = 0.30), the current use of wearable technology (*p* = 0.59), and the confidence in using wearable technology (*p* = 0.86).

### 3.3. Monitoring in the Field

A total of 70 out of 110 (63.64%) respondents preferred their health indicators monitored by both a SEMS operator and themselves while working in the field. Twenty-seven (27) (24.55%) preferred to monitor themselves only, nine (8.18%) preferred to have the SEMS operator be solely responsible, and four (3.64%) accounted for all other responses. Table 3 and Table 4 show the results of a univariate analysis that examined the association between the special operations group and age group and who should monitor their health while in the field. Both analyses show no significant difference in responses between the two groups.

A total of 71 out of 110 (63.64%) respondents preferred environmental indicators monitored by a SEMS operator and themselves while working in the field. Eighteen (18) (16.36%) preferred to have the SEMS operator be solely responsible, sixteen (16) (14.55%) preferred to monitor themselves only, and six (5.45%) accounted for all other responses. Table 3 and Table 4 show the results of the univariate analysis, and both show no significant difference in responses between the two groups.

### 3.4. Health Monitoring

The health indicators that respondents perceived as important for monitoring in the field are arranged in order of highest to lowest: heart rate (98.2%), blood pressure (93.7%), core body temperature (89.1%), hydration level (87.2%), and skin temperature (67.3%). Additionally, 87.4% selected their breathing rate to be monitored, 51.4% selected falls, and 48.6% selected stability. Of the respondents, 91.0% chose blood oxygen levels, 86.0% chose respiration carbon dioxide (CO_2_) levels, 86.5% selected cortisol levels (stress), 71.6% selected skin resistance, and 64.6% selected breathing depth levels to be monitored while working in the field. Table 5, Table 6, Table 7 and Table 8 reveal the results of the univariate analysis regarding health monitoring within the field. Cortisol levels (stress) (*p* = 0.02) and skin resistance and hydration levels (*p* = 0.05) resulted in significant correlations between the age of respondents. No other health indicators revealed any significant association between the special operations analysis and the age analysis. The analysis of special operations groups and heart rate could not determine significance due to low cell counts. Similarly, the analysis of age groups and heart rate as well as blood pressure could not determine significance due to low cell counts.

### 3.5. Environmental Monitoring

Respondents selected the following environmental agents/factors/attributes to be monitored: 100% selected CO, 99.1% H_2_S, 96.0% combustible gas, 95.4% O_2_, 88.4% particulates, 88.4% biological proteins, 87.3% CO_2_, 82.7% radiation, 78.6% ammonia, and 59.4% selected pH to be monitored while in the field. Additionally, 97.3% selected a lower exposure limit (LEL), 90.1% selected the temperature inside the suit, and 82.7% chose the temperature outside the suit to be monitored. Further, 69.7% selected humidity inside the suit, 68.8% selected humidity outside the suit, 53.2% chose noise levels inside the suit to be monitored, and 52.8% selected noise levels outside the suit. In addition, 98.2% chose hydrogen cyanide (HCN), 83.8% selected volatile organic compounds (VOCs), and 78.4% selected polyhalogenated compounds (PHCs) to be monitored while working in the field. Inside versus outside of the suit had an overlap of responses as it was the same respondents.

Table 9, Table 10, Table 11 and Table 12 present the results of the univariate analysis between the environmental indicators and the special operations analysis and the age analysis. Regarding special operations analysis, only biological proteins showed the significance of the association between the special operations and non-special operation groups (*p* = 0.03). Regarding age analysis, only carbon dioxide showed the significance of the association between age ≥ 42 and age < 42. Carbon monoxide could not determine significance due to all the respondents responding yes to the monitoring of carbon monoxide, and similar results were observed for the age analysis. The analysis of special operations groups could not determine the significance of hydrogen sulfide, combustible gas, lower explosive limit, and hydrogen cyanide due to low cell counts. The analysis of age groups could not determine the significance of hydrogen sulfide, lower explosive limit, and hydrogen cyanide due to low cell counts.

### 3.6. Additional Comments

The survey included three open-ended questions:Why respondents do not use wearable technology.Whether any additional types of health indicators should be monitored.Whether any other types of environmental hazards should be monitored.

Regarding wearable technology, sixteen respondents commented that current wearable devices were too expensive, and sixteen respondents said there was no need for them to wear them. One respondent commented, “I used to wear a watch while on duty for the more accurate taking of patients’ pulses. With the constant unknown of what type of call we would be going on, I would wear the watch all day every day. When we would get a call where we would need to put our bunker gear on, I would forget I was wearing the watch, and after so many times of putting the bunker gear on and working with it on, it would eventually break the bands on my watches. So, I quit wearing them as I was tired of replacing them.” Another commented that “… firefighting can easily damage wearable technology if it is not robust.”

As for health monitoring, five respondents would prefer blood glucose levels to be added as a physiological indicator. One respondent indicated that radon levels should be monitored at all stations, and one commented that wind direction and speed should be added as an environmental indicator that should be monitored.

## 4. Discussion

The overall goal of this exploratory study was to analyze the perceptions of first responders on using wearable technology and attitudes toward health and environmental monitoring. Based on current knowledge, this is the first survey to understand the views of first responders on wearable technology and health and environmental monitoring.

Our study identified five essential findings: (1) more than half (53%) of the respondents do not wear wearable technology for such reasons as they are expensive and break too easily, while 47% of respondents said yes to ever using wearable technology; (2) they prefer themselves and the SEMS operator to monitor their status while on shift; (3) the most essential health information to monitor was heart rate, blood pressure, cortisol levels, respiration carbon dioxide, blood oxygen saturation, respiratory rate, hydration level, and body temperature; (4) most of the environmental exposures first responders are exposed to are essential to be monitored, which led 100% of the respondents to select carbon monoxide to be monitored and 99% of the respondents to select hydrogen sulfide; (5) there was no significant difference between first responders that participated in special operations or non-special operations regarding the importance of health monitoring preferences; (6) there was a significant association between age group of the respondent and cortisol levels (stress) (*p* = 0.02) and a borderline significant association between age group of the respondent and skin resistance (*p* = 0.05); (7) there was a significant association between special operations and biological proteins (*p* = 0.03); and (8) there was a significant association between age group of the respondent and carbon dioxide (*p* = 0.03).

Environmental monitoring regardless of special operations vs. non-special operations or the age of the first responder was deemed important except for one variable, biological proteins. Special operations had a higher preference for monitoring biological proteins compared to non-special operations. Environmental monitoring regardless of age group (≥42 or <42) was also deemed important except for one variable, carbon dioxide. The age group < 42 had a higher preference for monitoring carbon dioxide compared to the ≥42 age group. There is no explanation for these observed differences. The sample size was small, which limits the power of the analysis. Age might be a confounding factor. There were also differences observed between the age of the respondent and two of the health variables. Age < 42 had a higher preference for monitoring stress and for monitoring skin resistance compared to those ≥42. A greater sample size would be needed as well as multivariable regression models to determine whether age and special operations were confounding variables. Overall, the high responses to environmental monitoring align with the concerns that first responders are exposed to toxic chemicals during fire suppression due to the increase in synthetic building materials and furnishing [23,24]. According to the Centers for Disease Control and Prevention (CDC), many of these chemicals, such as VOCs, radon, and HCN, are potential human carcinogens, and exposure is quite common among first responders. Even though first responders wear PPE during emergencies, dermal exposure still occurs. Skin exposure can occur through the first responder’s PPE through the hood, turnout jacket, and trousers [28]. Multiple studies have shown evaluated levels of chemical exposures on the skin following firefighting events [29,30,31,32]. Environmental monitoring also aligns with concerns about heat stress during events. Protective clothing worn by firefighters exacerbates heat exposure due to the limited ability to dissipate heat and moisture from the clothing microclimate to the external environment [33]. One study showed that three different prototype turnout suits significantly increased work time and significantly reduced core body temperature, skin temperature, and physiological strain compared to the control turnout suit [33]. Monitoring core temperature, external environment, and other markers of heat stress during an event may provide insight into areas where heat stress reduction can be improved whether it be protective clothing or other aspects of the job. The data from environmental and physiological monitoring can be used for more than health monitoring by examining areas of first responder duties that can be changed to improve health outcomes such as heat stress. Real-time monitoring of the environment will provide first responders with essential information when entering an incident that may have potential toxic gas exposure, such as fire suppression or hazmat emergencies. Further, it will provide additional information on how they can further protect themselves from exposures and improve health outcomes.

Cancer due to potential dermal exposure should not be the only worry for first responders. The CDC has stated that high levels of carbon monoxide (CO) in the environment can lead to CO poisoning, eventually leading to loss of consciousness, weakness, shortness of breath, or even death. Carbon dioxide (CO_2_) can lead to similar symptoms as CO, in addition to asphyxia, coma, and convulsions, according to The National Institution for Occupational Safety and Health (NIOSH). Another concern is hydrogen cyanide (HCN). According to OSHA, symptoms of HCN gas exposure include nausea, breathlessness, and headaches. All these gases mentioned (e.g., CO, CO_2_, and HCN) have been shown to exceed their respective short-term exposure limits on some emergency events for first responders [34].

Robustness and durability of wearable technology was a concern among respondents. Fitting sensing technology on clothes instead of a watch or band might be less obstructive and more durable [35]. Although biomonitoring sensors usually require accurate positioning of sensors, which may pose a problem for first responders due to the nature of duties encountered during events [35], the type of sensing technology for first responders is an area for further investigation. The cost might be a barrier for first responders as 16 respondents in our study cited current wearable devices as being too expensive as a reason preventing them from using wearable technology.

Interestingly, overall noise was not a significant concern for about half of the respondents in this study. However, other studies have shown first responders’ noise exposure limits exceed OSHA’s permissible exposure limits to noise during an eight-hour work shift [36,37]. Some first responders experience hearing loss early in their careers or accelerated hearing loss during their careers compared to other occupations [38,39]. Hearing is such a crucial part for first responders during emergencies since most of their communications and commands are verbally communicated. Additionally, the noise level has been shown to have a negative impact on CVD because it can increase blood pressure. A systematic review observed CVD in U.S. firefighters, which showed that for every 5-decibel increase in acute occupational noise, systolic blood pressure increases by 0.51 mm Hg. First responders can be exposed to nearly 90 decibels on average and may be exposed to as high as 166 decibels [14]. Possibly, the first responder community does not know the long-term effects of noise levels and health hazards that may follow due to hearing loss, and perhaps more outreach is needed to educate first responders on the long-term effects of hearing loss.

Other lesser concerns for first responders are falls and stability issues while working in the field. Even though, according to the National Fire Protection Association, in 2018, 22,975 injuries occurred at fire ground operations nearly 18% of these injuries were due to falls, slips, and jumps [40]. Perhaps first responders see these types of incidents as part of the job, or perhaps a false sense of security is occurring because of the PPE first responders wear while in the field.

### Limitations and Future Directions

As in any study, there are a couple of limitations. The first limitation is the generalizability of the study. All the participants were from one large metropolitan fire department consisting of 24 first stations. Further exploration should survey first responders in other locales so the study can be generalized. Another limitation is the small sample size; however, the response rate was good. A greater sample size would be able to better reduce type I and type II errors. Additionally, a multivariable analysis would be able to identify whether age or special operations were confounding factors. The observed significant differences from this study should be interpreted with caution. A possible extension of this study is surveying rural volunteer first responders, thus, providing another perspective. Additionally, other unmeasured factors could be explored with an expanded questionnaire. Although the study has limitations, it does provide valuable insights regarding first responders and their views on health and environmental monitoring while in the field.

## 5. Conclusions

In summary, no differences were found between special operations and non-special operations or the age of responders and their perceptions of wearable technology and monitoring their health and environment while in the field. While there were a few significant differences found between the age of responders and health and environmental variables. There was a significant difference between special operations groups for one of the environmental variables. This study did not have the power to assess the strength of association between the observed significant results and require further investigation. It is important to monitor the environment in order to prevent possible negative health effects since the environmental conditions have an impact on first responders’ health. Overall, our study illustrates that using and wearing monitoring devices is not age dependent, and that health and environmental concerns are important for first responders in any stage of their career. Special operations compared to non-special operations data shows that both are equally concerned about environmental agents and face the same health exposures. In addition, it can be concluded that the majority of respondents show an acceptance and willingness for monitoring their health and the environment while in the field.

## Figures and Tables

**Table 1 ijerph-20-04787-t001:** Univariate analysis of the association of special operations and wearable technology.

Wearable Technology	Special Operations N (%)	Non-Special Operations N (%)	X^2^ (df, N) = [X^2^ Value], *p* = [*p* Value]
Ever Used Wearable Technology			X^2^ (1, 112) = 2.59, *p* = 0.11
Yes	20 (17.86)	33 (29.46)	
No	14 (12.50)	45 (40.18)	
Currently Use Wearable Technology			X^2^ (1, 112) = 0.03, *p* = 0.86
Yes	12 (22.64)	19 (35.85)	
No	8 (15.09)	14 (26.42)	
How confident are you using Wearable Technology			X^2^ (1, 112) = 2.6, *p* = 0.11
High confidence	4 (12.90)	12 (38.17)	
Low confidence	8 (25.81)	7 (22.58)	

**Table 2 ijerph-20-04787-t002:** Univariate analysis of the association between the age of the respondents and wearable technology.

Wearable Technology	Age ≥ 42 N (%)	Age < 42 N (%)	X^2^ (df, N) = [X^2^ Value], *p* = [*p* Value]
Ever Used Wearable Technology			X^2^ (1, 112) = 1.10, *p* = 0.30
Yes	29 (25.89)	24 (21.43)	
No	38 (33.93)	21 (25.89)	
Currently Use Wearable Technology			X^2^ (1, 112) = 0.29, *p* = 0.59
Yes	16 (30.19)	15 (28.30)	
No	13 (24.53)	9 (16.98)	
How confident are you using Wearable Technology			X^2^ (1, 112) = 0.03, *p* = 0.86
High confidence	8 (25.81)	8 (25.81)	
Low confidence	5 (16.13)	7 (22.58)	

**Table 3 ijerph-20-04787-t003:** Univariate analysis of the association between special operations and who should monitor health and environment.

Monitoring	Special Operations N (%)	Non-Special Operations N (%)	X^2^ (df, N) = [X^2^ Value], *p* = [*p* Value]
Who Should Monitoring Your Health while working in the Field			X^2^ (3, 112) = 0.31, *p* = 0.96
SEMS Operator and myself	20 (18.8)	50 (45.45)	
SEMS Operator only	3 (2.73)	6 (5.45)	
Myself	9 (8.18)	18 (16.36)	
Other	1 (0.91)	3 (2.73)	
Who Should Monitoring Your Environment while working in the Field			X^2^ (3, 112) = 0.86, *p* = 0.83
SEMS Operator and myself	22 (20.0)	48 (43.64)	
SEMS Operator only	6 (5.45)	12 (10.91)	
Myself	4 (3.64)	12 (10.91)	
Other	1 (0.91)	5 (4.55)	

**Table 4 ijerph-20-04787-t004:** Univariate analysis of the association between the age of respondents and who should monitor health and environment.

Monitoring	Age ≥ 42 N (%)	Age < 42 N (%)	X^2^ (df, N) = [X^2^ Value], *p* = [*p* Value]
Who Should Monitoring Your Health while working in the Field			X^2^ (3, 112) = 5.22, *p* = 0.16
SEMS Operator and myself	40 (36.36)	30 (27.27)	
SEMS Operator only	3 (2.73)	6 (5.45)	
Myself	20 (18.18)	7 (6.36)	
Other	2 (1.82)	2 (1.82)	
Who Should Monitoring Your Environment while working in the Field			X^2^ (3, 112) = 5.06, *p* = 0.17
SEMS Operator and myself	41 (37.27)	29 (26.36)	
SEMS Operator only	8 (7.27)	10 (9.09)	
Myself	13 (11.82)	3 (2.73)	
Other	3 (2.73)	3 (2.73)	

**Table 5 ijerph-20-04787-t005:** Univariate analysis of the association between special operations and health monitoring (subgroup a).

Health Monitoring	Special Operation N (%)	Non-Special Operations N (%)	X^2^ (df, N) = [X^2^ Value], *p* = [*p* Value]
Heart Rate			-
Yes	34 (30.63)	75 (67.57)	
No	0 (0.00)	2 (1.80)	
Blood Pressure			X^2^ (1, 112) = 0.53, *p* = 0.47
Yes	31 (27.93)	73 (65.77)	
No	3 (2.70)	4 (3.60)	
Core Body Temperature			X^2^ (1, 112) = 3.21, *p* = 0.07
Yes	33 (30.00)	65 (59.09)	
No	1 (0.91)	11 (10.00)	
Skin Temperature			X^2^ (1, 112) = 2.44, *p* = 0.12
Yes	25 (23.36)	47 (43.93)	
No	7 (6.54)	28 (26.17)	
Hydration Level			X^2^ (1, 112) = 1.95, *p* = 0.16
Yes	31 (28.44)	64 (58.72)	
No	2 (1.83)	12 (11.01)	
Stability			X^2^ (2, 112) = 0.89, *p* = 0.64
Yes	38 (34.86)	15 (13.76)	
No	14 (12.84)	9 (8.26)	
Don’t Know	23 (21.10)	10 (9.17)	
Falls			X^2^ (2, 112) = 2.02, *p* = 0.36
Yes	16 (14.68)	40 (36.70)	
No	11 (10.09)	15 (13.76)	
Don’t Know	7 (6.42)	20 (18.35)	

(-) No significance can be determined.

**Table 6 ijerph-20-04787-t006:** Univariate analysis of the association between special operations and health monitoring (subgroup b).

Health Monitoring	Special Operations N (%)	Non-Special Operations N (%)	X^2^ (df, N) = [X^2^ Value], *p* = [*p* Value]
Breathing Rate			X^2^ (1, 112) = 1.13, *p* = 0.29
Yes	28 (25.23)	69 (62.16)	
No	6 (5.41)	8 (7.21)	
Breathing Depth			X^2^ (1, 112) = 0.17, *p* = 0.68
Yes	21 (19.09)	50 (45.45)	
No	13 (11.82)	26 (23.64)	
Blood Oxygen Levels			X^2^ (1, 112) = 0.45, *p* = 0.50
Yes	30 (27.03)	71 (63.96)	
No	4 (3.60)	6 (5.41)	
Respiration CO_2_ Levels			X^2^ (1, 112) = 0.21, *p* = 0.65
Yes	30 (28.04)	62 (57.94)	
No	4 (3.74)	11 (10.28)	
Cortisol Levels (Stress)			X^2^ (1, 112) = 0.13, *p* = 0.72
Yes	30 (27.03)	66 (59.46)	
No	4 (3.60)	11 (9.91)	
Skin Resistance and Hydration Levels			X^2^ (1, 112) = 2.45, *p* = 0.12
Yes	27 (24.77)	51 (46.79)	
No	6 (5.50)	25 (22.94)	

**Table 7 ijerph-20-04787-t007:** Univariate analysis of the association between the age of the respondents and health monitoring (subgroup a).

Health Monitoring	Age ≥ 42 N (%)	Age < 42 N (%)	X^2^ (df, N) = [X^2^ Value], *p* = [*p* Value]
Heart Rate			-
Yes	65 (58.56)	44 (39.64)	
No	2 (1.80)	0 (0.00)	
Blood Pressure			-
Yes	61 (54.95)	43 (38.74)	
No	6 (5.41)	1 (0.90)	
Core Body Temperature			X^2^ (1, 112) = 0.02, *p* = 0.90
Yes	59 (53.84)	39 (35.45)	
No	7 (6.36)	5 (4.55)	
Skin Temperature			X^2^ (1, 112) = 0.20, *p* = 0.65
Yes	42 (39.25)	30 (28.04)	
No	22 (20.56)	13 (12.15)	
Hydration Level			X^2^ (1, 112) = 0.80, *p* = 0.37
Yes	56 (51.38)	39 (35.78)	
No	10 (9.17)	4 (3.67)	
Stability			X^2^ (2, 112) = 2.02, *p* = 0.36
Yes	32 (29.36)	21 (19.27)	
No	11 (10.09)	12 (11.01)	
Don’t Know	22 (20.18)	11 (10.09)	
Falls			X^2^ (2, 112) = 1.51, *p* = 0.47
Yes	36 (33.03)	20 (18.35)	
No	13 (11.93)	13 (11.93)	
Don’t Know	16 (14.68)	11 (10.09)	

(-) No significance can be determined.

**Table 8 ijerph-20-04787-t008:** Univariate analysis of the association between the age of the respondents and health monitoring (subgroup b).

Health Monitoring	Age ≥ 42 N (%)	Age < 42 N (%)	X^2^ (df,N) = [X^2^ Value], *p* = [*p* Value]
Breathing Rate			X^2^ (1, 112) = 0.10, *p* = 0.75
Yes	58 (52.25)	39 (35.14)	
No	9 (8.11)	5 (4.50)	
Breathing Depth			X^2^ (1, 112) = 0.95, *p* = 0.33
Yes	45 (40.91)	26 (23.64)	
No	21 (19.09)	18 (16.36)	
Blood Oxygen Levels			X^2^ (1, 112) = 0.41, *p* = 0.52
Yes	61 (54.95)	40 (36.04)	
No	5 (4.50)	5 (4.50)	
Respiration CO_2_ Levels			X^2^ (1, 112) = 0, *p* = 0.99
Yes	55 (51.40)	37 (34.58)	
No	9 (8.41)	6 (5.61)	
Cortisol Levels (Stress)			X^2^ (1, 112) = 5.33, *p* = 0.02 **
Yes	53 (47.75)	43 (38.74)	
No	13 (11.71)	2 (1.80)	
Skin Resistance and Hydration Levels			X^2^ (1, 112) = 3.82, *p* = 0.05 *
Yes	42 (38.53)	36 (33.03)	
No	23 (21.10)	8 (7.34)	

** Statistically significant result at *p* < 0.05. * Borderline significant result.

**Table 9 ijerph-20-04787-t009:** Univariate analysis of the association between special operations and environmental monitoring (subgroup a).

Environmental Monitoring	Special Operations N (%)	Non-Special Operations N (%)	X^2^ (df, N) = [X^2^ Value], *p* = [*p* Value]
pH			X^2^ (1, 112) = 3.52, *p* = 0.06
Yes	24 (22.64)	39 (36.79)	
No	9 (8.49)	34 (32.08)	
Oxygen			X^2^ (1, 112) = 0.31, *p* = 0.58
Yes	33 (30.28)	71 (65.14)	
No	1 (0.92)	4 (3.67)	
Carbon Monoxide			-
Yes	34 (30.91)	76 (69.09)	
No	0 (0.00)	0 (0.00)	
Hydrogen Sulfide			-
Yes	33 (29.46)	78 (69.64)	
No	1 (0.89)	0 (0.00)	
Combustible Gas			-
Yes	34 (30.36)	74 (66.07)	
No	0 (0.00)	4 (3.57)	
Ammonia			X^2^ (1, 112) = 0.41, *p* = 0.52
Yes	28 (25.00)	60 (53.57)	
No	6 (5.36)	18 (16.07)	
Particulates			X^2^ (1, 112) = 0.37, *p* = 0.54
Yes	31 (27.68)	68 (60.71)	
No	3 (2.68)	10 (8.93)	
Carbon Dioxide			X^2^ (1, 112) = 2.74, *p* = 0.10
Yes	27 (24.55)	69 (62.73)	
No	7 (6.36)	7 (6.36)	
Biological Proteins			X^2^ (1, 112) = 4.94, *p* = 0.03 **
Yes	29 (26.36)	49 (44.55)	
No	5 (4.55)	27 (24.55)	
Radiation			X^2^ (1, 112) = 1.04, *p* = 0.31
Yes	30 (27.27)	61 (55.45)	
No	4 (3.64)	15 (13.64)	
LEL (Lower Explosive Limit)			-
Yes	32 (29.36)	74 (67.89)	
No	1 (0.92)	2 (1.83)	

(-) No significance can be determined. ** Statistically significant result at *p* < 0.05.

**Table 10 ijerph-20-04787-t010:** Univariate analysis of the association between special operations and environmental monitoring (subgroup b).

Environmental Monitoring	Special Operations N (%)	Non-Special Operations N (%)	X^2^ (df, N) = [X^2^ Value], *p* = [*p* Value]
Temperature Inside the Suit			X^2^ (1, 112) = 0.89, *p* = 0.35
Yes	32 (28.33)	68 (61.26)	
No	2 (1.80)	9 (8.11)	
Temperature Outside the Suit			X^2^ (1, 112) = 2.46, *p* = 0.12
Yes	31 (28.18)	60 (54.55)	
No	3 (2.73)	16 (14.55)	
Humidity Inside the Suit			X^2^ (1, 112) = 1.07, *p* = 0.30
Yes	26 (23.85)	50 (45.87)	
No	8 (7.34)	25 (22.94)	
Humidity Outside the Suit			X^2^ (1, 112) = 1.35, *p* = 0.24
Yes	26 (23.85)	49 (44.95)	
No	8 (7.34)	26 (23.85)	
Noise Inside the Suit			X^2^ (2, 112) = 0.03, *p* = 0.99
Yes	18 (16.51)	40 (36.70)	
No	10 (9.17)	21 (19.27)	
Don’t Know	6 (5.50)	14 (12.84)	
Noise Outside the Suit			X^2^ (2, 112) = 0, *p* = 1
Yes	18 (16.67)	39 (36.11)	
No	10 (9.26)	22 (20.37)	
Don’t Know	6 (5.56)	13 (12.04)	
Hydrogen Cyanide			-
Yes	33 (29.73)	76 (68.47)	
No	1 (0.90)	1 (0.90)	
Volatile Organic Compound			X^2^ (1, 112) = 1.97, *p* = 0.16
Yes	31 (27.93)	62 (55.86)	
No	3 (2.70)	15 (13.51)	
Polyhalogenated Compound			X^2^ (1, 112) = 0.03, *p* = 0.86
Yes	27 (24.32)	60 (54.05)	
No	7 (6.31)	17 (15.32)	

(-) No significance can be determined.

**Table 11 ijerph-20-04787-t011:** Univariate analysis of the association between the age of the respondents and environmental monitoring (subgroup a).

Environmental Monitoring	Age ≥ 42 N (%)	Age < 42 N (%)	X^2^ (df, N) = [X^2^ Value], *p* = [*p* Value]
pH			X^2^ (1, 112) = 0.03, *p* = 0.86
Yes	37 (34.91)	26 (24.53)	
No	26 (24.53)	17 (16.04)	
Oxygen			X^2^ (1, 112) = 0.90, *p* = 0.34
Yes	61 (55.96)	43 (39.45)	
No	4 (3.67)	1 (0.92)	
Carbon Monoxide			-
Yes	66 (60.00)	44 (40.00)	
No	0 (0.00)	0 (0.00)	-
Hydrogen Sulfide			
Yes	67 (59.82)	44 (39.29)	
No	0 (0.00)	1 (0.89)	
Combustible Gas			X^2^ (1, 112) = 0.17, *p* = 0.68
Yes	65 (58.04)	43 (38.39)	
No	2 (1.79)	2 (1.79)	
Ammonia			X^2^ (1, 112) = 0.09, *p* = 0.76
Yes	52 (46.43)	36 (32.14)	
No	15 (13.39)	9 (8.04)	
Particulates			X^2^ (1, 112) = 1.79, *p* = 0.18
Yes	57 (50.89)	42 (37.50)	
No	10 (8.93)	3 (2.68)	
Carbon Dioxide			X^2^ (1, 112) = 4.70, *p* = 0.03 **
Yes	53 (48.18)	43 (39.09)	
No	12 (10.91)	2 (1.82)	
Biological Proteins			X^2^ (1, 112) = 1.74, *p* = 0.19
Yes	43 (39.09)	35 (31.82)	
No	22 (20.00)	10 (9.09)	
Radiation			X^2^ (1, 112) = 0.16, *p* = 0.69
Yes	53 (48.18)	38 (34.55)	
No	12 (10.91)	7 (6.36)	
LEL (Lower Explosive Limit)			-
Yes	64 (58.72)	42 (38.53)	
No	1 (0.92)	2 (1.83)	

(-) No significance can be determined. ** Statistically significant result at *p* < 0.05.

**Table 12 ijerph-20-04787-t012:** Univariate analysis of the association between the age of the respondents and environmental monitoring (subgroup b).

Environmental Monitoring	Age ≥ 42 N (%)	Age < 42 N (%)	X^2^ (df, N) = [X^2^ Value], *p* = [*p* Value]
Temperature Inside the Suit			X^2^ (1, 112) = 0.05, *p* = 0.82
Yes	60 (54.05)	40 (36.04)	
No	7 (6.31)	4 (3.60)	
Temperature Outside the Suit			X^2^ (1, 112) = 0.04, *p* = 0.84
Yes	55 (50.00)	36 (32.73)	
No	11 (10.00)	8 (7.27)	
Humidity Inside the Suit			X^2^ (1, 112) = 0.08, *p* = 0.78
Yes	46 (42.40)	30 (27.52)	
No	19 (17.43)	14 (12.84)	
Humidity Outside the Suit			X^2^ (1, 112) = 0.09, *p* = 0.76
Yes	44 (40.37)	31 (28.44)	
No	21 (19.27)	13 (11.93)	
Noise Inside the Suit			X^2^ (2, 112) = 0.35, *p* = 0.84
Yes	36 (33.03)	22 (20.18)	
No	18 (16.51)	13 (11.99)	
Don’t Know	11 (10.09)	9 (8.26)	
Noise Outside the Suit			X^2^ (2, 112) = 1.65, *p* = 0.44
Yes	34 (31.48)	23 (21.30)	
No	21 (19.44)	11 (10.19)	
Don’t Know	9 (8.33)	10 (9.26)	
Hydrogen Cyanide			-
Yes	66 (59.46)	43 (38.74)	
No	1 (0.90)	1 (0.90)	
Volatile Organic Compound			X^2^ (1, 112) = 1.26, *p* = 0.26
Yes	54 (48.65)	39 (35.14)	
No	13 (11.71)	5 (4.50)	
Polyhalogenated Compound			X^2^ (1, 112) = 0.51, *p* = 0.48
Yes	51 (45.95)	36 (32.43)	
No	16 (14.41)	8 (7.21)	

(-) No significance can be determined.

## Data Availability

Data available on request due to restrictions e.g. privacy or ethical. The data presented in this study are available on request from the corresponding author. The data are not publicly available due to University of Nebraska Medical Center Institutional Review Board restrictions.

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
