# Peer review of "Exploring First Responders’ Use and Perceptions on Continuous Health and Environmental Monitoring"

_ijerph, 2023, doi:10.3390/ijerph20064787_

Round 1
Reviewer 1 Report
The manuscript by Dr. Grothe and coworkers used the survey data on attitudes towards health and environmental monitoring to determine which health and environmental indicators should be monitored first responders worked in fire departments in Omaha, Nebraska. This is an important issue; however, there are several major and minor flaws in this study.
Major:
1. The authors should update the latest references. No reference from the 2020-2022 has been cited in the paper. It means that no researchers studied on the attitudes towards health and environmental monitoring for first responders. Please add and clarify more detail in the text.
2. In the Materials and Methods section, (1) Page 4, lines 198-200; Page 5, lines 203-210: Any related references of studies to the question measures should be cited. (2) Fisher's exact test is only used to analyze 2×2 contingency tables. The chi-square test examines whether rows and columns (e.g., 2×3 or 3×2) of a contingency table. However, the chi-square test cannot be used when more than 20% of the expected frequencies are under 5. The authors need to add more information in the text and check the results.
3. In the Results section, (1) In Table 1-12, the authors need to report exact P values for these analyses, not just "P>0.05" or "P=0.05". Moreover, the p-values were not used for Fisher's exact test. Please revise and add in the text. (2) In Table 6, 8, 10, and 12, why did the authors add "(continued)" to title of tables? This made me confused. If the contents of the tables (Table 5 and Table 6; Table 7 and Table 8; Table 9 and Table 10; Table 11 and Table 12) continue, both table descriptions have the same number of table elements. Please check and modify it.
4. In the Discussion section, (1) Unmeasured factors, such as work seniority, working hours per week, and the average exposure per unit of time of respondents may also influence the results in attitudes towards health and environmental monitoring. Please make clear in the limitations of the present study in the text. (2) There were no significant differences in distributions of health and environmental monitoring between special and non-special operations or the age of responders. The authors need to explain practical implications and suggest directions for future research based on the findings of this study, respectively. Please explain and add the relevant details in the text.
5. In the Conclusion section, the description is too general and simplistic, especially no significant differences of outcomes between first responders. In addition, the diverse assessment of health and environmental risk factors need to be explained, respectively. Please clarify and modify this more detail in the text.
6. The ‘citation formatting’ of several references (Ref. 1-4, 6, 8, 11, 13, 16, 17, 19, 22-24, 26, 30, 31, 33, 36, 39) is inaccuracy and inconsistent, for example the article title, URL ("web address"), etc.. Please check reference formatting in detail according to the instructions of the Information for Authors of International Journal of Environmental Research and Public Health.
Minor:
1. The order of citation numbers in the text is not sequential. Please check and modify it.
2. Page 7, lines 291-292: In Table 3, the title of the table should move to the top of the table. Please modify it.
3. The abbreviation (i.e., SEMS) should be defined in full words and appeared in the note below (i.e., SEMS, Safety and Environmental Management System) the Table 3 and Table 4. In addition, the authors have to check and correct the information about “…Paramedics, Captain, Safety Officer, Personality Accountability System (SEMS) operator… (Page 5, lines 205-206)”.
Author Response
Please see the attachment of our comprehensive response to reviewers.

Reviewer 2 Report
A study was conducted to identify the acceptance of first responders towards continuous health and environmental monitoring through wearable devices. This study is worthy of investigation and possesses value as a key reference to improve the health and safety of first responders. However, there are a few concerns. Mainly, the rather small sample size and unequal distribution of groups might influence the authenticity of the results and findings from this research. There are also areas in which this study can further improve. The specific comments are attached.

Author Response

(The authors gave the same response as above.)
